# Acquisition of the physiological quality of peanut (*Arachis hypogaea* L.) seeds during maturation under the influence of the maternal environment

**Maurício Hideki Okada**[☯]**, Gustavo Roberto Fonseca de Oliveira**[☯]**, Maria Márcia Pereira Sartori**[‡]**, Carlos Alexandre Costa Crusciol** [iD][‡]**, João Nakagawa**[‡]**, Edvaldo Aparecido Amaral da Silva** [iD]*[☯]

Crop Production Department, Universidade Estadual Paulista, Botucatu, São Paulo, Brazil

☯ These authors contributed equally to this work.
‡ MMPS, CACC and JN also contributed equally to this work.
* amaral.silva@unesp.br

**Data Availability Statement:** All relevant data are within the paper and its Supporting Information files.

## Abstract

The scarcity of information on the maturation physiology of the peanut seed (*Arachis hypogaea* L.; Virgínia group) makes harvesting high quality seeds a challenge for the seed industry. During two consecutive crop seasons, we studied the acquisition of physiological quality of peanut seeds during maturation in tropical conditions. We bring new insights about the period of late maturation of seeds and the influence of the maternal environment on physiological quality. We monitored water content, dry weight, ability of germination, desiccation tolerance, vigor and longevity. In addition, we monitored temperature and precipitation throughout plant growth. We demonstrate that the physiological quality of peanut seeds is acquired during development, with a maximum between 57 and 76 days after flowering in the late stage of maturation. This final period represents about 25% of the development, considered the best time to harvest peanut seeds with the highest quality. Our findings also support the idea that the adequate proportion of rainfall and thermal sum in the maternal environment are factors that favor the acquisition of peanut seed longevity.

## Introduction

Peanuts (*Arachis hypogaea* L.) belong to the Fabaceae family and are among the five most important oil species in the world [1]. The estimated production for 2020/21 is about 47.79 million tons [2], of which 65% are from China, India, Nigeria and the United States [3]. Peanuts are a source of protein and oil for human and animal nutrition [4] and their grains are consumed by nations of the six continents [3]. This context highlights the need for research regarding factors that interfere in the production of the species, which has contributed to food security and to the global economy [1, 5]. The factor that we highlight is the physiological quality of the seeds, an essential requirement for the production of cultivated plants [6] and for the achievement of high productivity [7]. It also expresses the seed storage potential as a strategic

**Funding:** This study was funded by: National Council for Scientific and Technological Development 130193 / 2017-8 for M.H. Okada, 142236 / 2020-9 for G.R.F. Oliveira and 309718 / 2018-0 for E.A. Amaral da Silva.

**Competing interests:** The authors have declared that no competing interests exist.

attribute for conservation of biodiversity and agricultural activity [6, 8]. However, we still know little about the attributes involved in the acquisition of the physiological quality of peanut seeds and when they are acquired. One strategy to fill this gap is to study the acquisition of quality in peanut seeds during maturation, which has been reported for legumes such as soybean [9, 10], *Medicago truncatula* [11, 12] and cowpea beans [13].

The flowering of *Arachis hypogaea* plants is characterized as indeterminate, resulting in the prolonged emission of pegs at different times during the reproductive phases [14, 15]. The consequence of this characteristic is the continuous formation of pods and seeds, which normally present multiple stages of development at the end of the crop cycle [1]. Thus, during harvest, a large proportion of seeds did not complete their development and as a consequence, a significant part of the seed lot that is intended to be marketed will not be of high quality [16, 17]. Little information is available on the physiology of peanut seeds during maturation. This reinforces the need of studies to clarify an important question for peanut producers around the world: at what point in time of development can peanut seeds be harvested with the highest physiological quality? The answer to this question comes from understanding the physiological events involved during the maturation and late maturation of peanut seeds, stages in which the ability of seed germination, desiccation tolerance, vigor and longevity are sequentially acquired [18].

It is known that immature seeds still haven't fully acquired the aforementioned attributes of physiological quality [10, 19]. This means that the preparatory mechanisms for the dry state and maximum longevity can be impaired, since they are mainly acquired in the late maturation of orthodox seeds [9, 20]. From a practical point of view, immature seeds have a shorter life span [8], which make it difficult to maintain quality standards during the storage period. Therefore, we believe that knowing the physiological events that happen during maturation and late maturation and the identification of the moment when most of the peanut seeds have acquired superior quality will bring benefits to the seed industry. This means that seeds can be commercialized with superior quality and hence comply with the requirements established by the regulatory agencies.

The consensus about the harvesting moment can vary between groups of cultivated species [21], and the current knowledge on the subject reinforces the idea that the maximum accumulation of dry weight (mass maturity) does not mean the end of the acquisition of seed quality [18, 20]. As an example, we can mention what happens with okra seeds (*Abelmoschus esculentus*), which after the seed filling phase, take around 20 days to reach maximum physiological quality [22]. In the case of soybean (*Glycine max*), this period is around 10 days [10]. The late maturation phase, although neglected in many species, including peanuts, is decisive for the acquisition of physiological quality [21]. We emphasize that in this final period of development, the acquisition of longevity in leguminous seeds is sensitive to factors in the maternal environment such as temperature [23] and water availability [11], which can alter the maximum seed quality. In the case of peanuts, these relationships still need to be understood.

In view of these considerations, we emphasize in our hypothesis that peanut seeds acquire physiological quality throughout maturation and late maturation, reaching maximum quality after mass maturity. Our expectation also reinforces the idea that climatic variables such as thermal sum and water availability in the maternal environment, when in an adequate proportion throughout plant growth, favor seed longevity. If we can validate these hypotheses, our findings will promote an understanding of the best time to harvest peanut seeds with superior quality, as well as a clarification of whether or not this point coincides with the maximum accumulation of dry mass. Therefore, our aim over two consecutive crop seasons was to monitor the acquisition of the physiological quality of peanut seeds during maturation and late maturation under tropical conditions.

## Material and methods

### Characterization of the seed production area

Peanut seeds (*Arachis hypogaea* L., Virgínia group) were produced in the commercial seed production area belonging to the city of Sertãozinho, SP / Brazil, for two consecutive crop seasons during the years of 2017 and 2018. The soil was characterized as an Oxisol [24], with a history of cultivation with sugar cane for five years. The area was subjected to soil preparation (plowing and harrowing), liming and sowing fertilization (triple superphosphate; 46% $P_2O_5$) according to the chemical attributes of the soil collected from the 0.0 to 20.0 cm layer (pH $CaCl_2$: 5.1; Organic matter: 25.0 g $dm^{-3}$; P resin: 7.0 mg $dm^{-3}$; S: 7.0 mg $dm^{-3}$; $Al^{3+}$: 0.0 mmolc $dm^{-3}$; $H+Al^{3+}$: 28.0 mmolc $dm^{-3}$; K: 1.4 mmolc $dm^{-3}$; Ca: 27.0 mmolc $dm^{-3}$; Mg: 5.0 mmolc $dm^{-3}$; sum of bases: 33.0 mmolc $dm^{-3}$; CTC: 62 mmolc $dm^{-3}$; base saturation: 54% and Al saturation: 0%). Before sowing, the seeds were treated with a commercial fungicide (active ingredients: carboxim and tiram; applied dose: 2 mL $kg^{-1}$) and were mechanically sown at a depth of 5 cm and a spacing of 90 cm. The sowing density aimed at a stand of 15 plants per meter.

### Characterization of the reproductive stages

The flowers were tagged to identify the beginning of the opening of the floral bud and subsequently to monitor the number of days after flowering (DAF). Rainfall and air temperature (maximum and minimum) of the production environment were monitored from sowing to harvest of the peanut seeds (S1 Fig). The seeds were manually harvested; the fruits were washed under running water and dried at room temperature. The seeds were manually extracted from the fruits and classified according to the reproductive stages [25] which are R5, R6, R7, R8 and R9, corresponding to 28, 35, 43, 57 and 76 DAF, respectively. The characterization of the seed quality (water content, dry weight, ability of germination, desiccation tolerance, vigor and longevity) was performed for each reproductive stage. Then, photos of the seeds and pods were taken (digital camera model CannonSX50HS) to characterize the stages of seed and fruit development (Fig 1).

### Characterization of seed quality

The water content was determined by the oven method at 105 ± 3° C for 24 hours, using 4 repetitions of 20 seeds each [26]. To evaluate the dry weight of the seeds, during seed

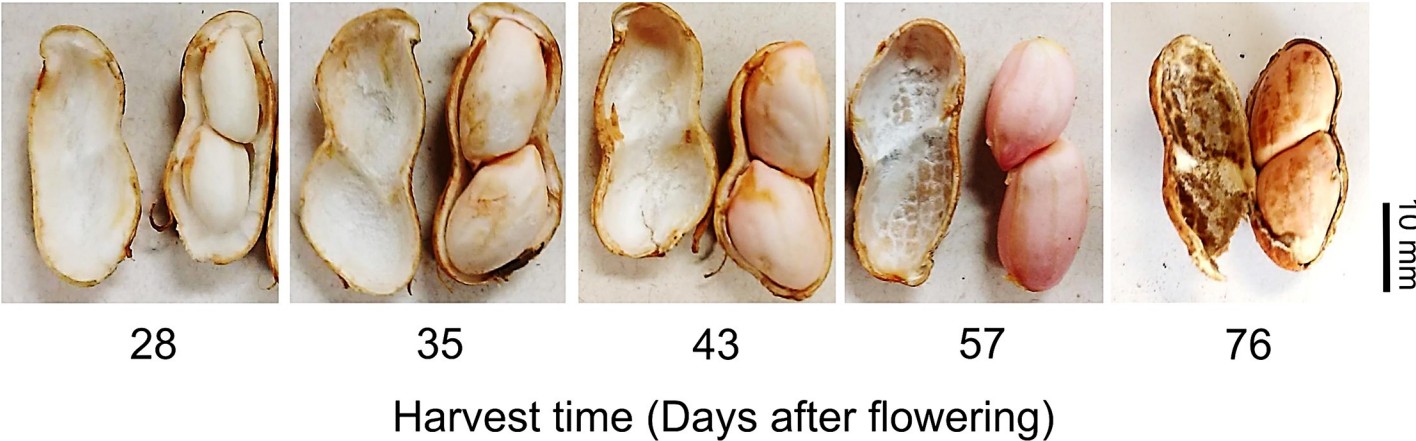

28          35          43          57          76

## Harvest time (Days after flowering)

**Fig 1. Seed and fruit development of *Arachis hipogaea* L. (Virgínia group).** Phenological stages during the acquisition of seed physiological quality harvested in the crop season 2017 at 28 (R5), 35 (R6), 43 (R7), 57 (R8) and 76 (R9) days after flowering (DAF), respectively. 76 days after flowering (R9 stage) represents the mature seeds and there is no connection of the seed with the fruit tissue.

development, four replications of 20 seeds were weighed, on an analytical scale with an accuracy of 0.001g, after drying at 60° C for 72 h. The results were expressed in grams of water per gram of dry weight. Seeds were treated with sodium hypochlorite (1% for 10 min) for disinfection before the physiological tests and to break any residual primary dormancy, as described for *Brassica tournefortii* [27].

For the germination test, we used four replicates of 25 seeds which were placed between rolled paper, moistened with deionized water, and kept at a constant temperature of 25° C in the dark. The number of germinated seeds was determined daily, and the seeds were considered germinated when they presented a primary root with a length ≥ 2 mm.

Desiccation tolerance was tested by drying the seeds in an environment with 42% relative humidity (RH), generated by a saturated saline solution of $K_2CO_3$, at a temperature of 25° C [28]. The drying time varied from 24 to 72 hours. After drying, the seeds were pre-humidified in an environment of 95–100% RH, at 25° C for 24 hours. Then, the seeds were placed to germinate as described above. The seeds that formed normal seedlings after drying were considered tolerant to desiccation.

During the germination test, we also evaluated seed vigor by determining the time required for 50% of the seeds to germinate (t50), measured in hours and with radicle protrusion as the criterion [29]. In parallel, we also measured seed vigor by counting the number of germinated seeds at five days after the beginning of the germination test, with the percentage of normal seedlings as the criterion.

For the study of longevity, the seeds from each reproductive stage were placed in a hermetically sealed box, with 75% RH at 35° C, containing a saturated NaCl solution (40 mL of distilled water and 18 g of salt). RH and temperature were monitored using a data logger. The seeds were sampled at intervals of seven days. Longevity was determined considering the storage time necessary to reduce the germination of the seed population to 50% (p50) (S3 Table), derived from sigmoid survival curves (f = a/(1+exp(-(x-x0)/b))), which express the ability of germination percentage over storage time (75% RH, 35° C). The data were transformed into probit to determine p50 by using the equation: v = (Ki-p)/σ, where: v = viability in days, Ki = initial germination in probit values, p = expected death over time and σ = slope of the curve [30].

Duration of the three developing phases of peanut seed was calculated based on the proportion of each stage (in days) in relation to the total time of seed development. The values were expressed as a percentage. The seeds started to be harvested at 28 DAF, which is considered the beginning of the maturation phase. The harvesting continued up to 76 DAF, which is considered the end of the late maturation phase.

## Calculation of thermal sum (Degrees-Days) and rainfall during seed production

The thermal sum was determined by monitoring temperatures during the two crop seasons, from sowing to harvest. The calculation was performed based on Degrees-Days (DD) [31]. The lowest base temperature (Tb) used was 10° C, which is used for most crops. The upper base temperature (TB) was 33° C, adopted for peanut under tropical conditions [32]. The water availability accumulated during seed production was quantified through the sum of precipitation data observed in the seed production areas from sowing to harvest in the years of 2017 and 2018.

## Statistical analysis

The normality of the residues was evaluated (Shapiro-Wilk test; software R core Team; package 'ExpDes.pt') from the data observed for each variable studied. The data were transformed

(Box Cox transformation) as necessary to meet the assumptions of the analysis of variance (ANOVA). We used a completely randomized experimental design, with five maturation stages (R5, R6, R7, R8 and R9) as a source of variation with four replications for each stage with 20 seed per replication. The data obtained at each reproductive stage were subjected to ANOVA and the LSD test at a significance level of 5%. The variables were analyzed separately for each crop season in the years of 2017 and 2018 (S1 and S2 Tables). In the two years of study, the water content, dry weight, seed physiological quality, degree days (thermal sum) and rainfall were subjected to analysis of principal components. For the respective analyses, we used the Permanova test and the Bray-Curtis similarity index to identify significance between the groups obtained according to the maturation stages at a significance level of 1% (software Canoco 5).

## Results

The initial water content of the peanut seeds was maximum (around 70 g) at 28 DAF and gradually reduced until the end of the seed development phase at 76 DAF (Figs 2A and 3A). In parallel, there was a progressive accumulation of dry weight up to 57 DAF (mass maturity) and then remained stable until 76 DAF (late maturation phase) when the seeds were harvested. The seeds started to germinate at 28 DAF, while the beginning of the acquisition of desiccation tolerance occurred at 35 DAF. Regardless of the agricultural year, the ability of germination and desiccation tolerance reached a maximum percentage at 57 DAF and remained stable at 76 DAF (Figs 2B and 3B).

Seed vigor increased between 35 and 43 DAF according to the t50 test, when 50% of the seeds had germinated (radicle protrusion). The ability of seeds to form normal seedlings was manifested from 35 DAF onwards. Peanut seeds acquired maximum vigor at the end of development (late maturation) at 76 DAF, regardless of the test used and the agricultural year studied (Figs 2C and 3C).

The ability of peanut seeds to tolerate storage (35˚C and 75% RH) started at 43 DAF in the two years of study; seeds collected at 28 and 35 DAF lost viability in less than ten days of storage (Fig 4A and 4B). Seed longevity, determined by the p50 test, was the last attribute of physiological quality to be acquired and was maximum at 76 DAF in the late maturation phase (Fig 5A and 5B).

Principal component analysis showed the influence of the maternal environment on the acquisition of seed longevity (Fig 6). According to the Permanova test, there was a significant difference (at a significance level of 1%; $p < 0.01$) between the variables observed at each stage of maturation. The first component explained 92.6% of the variables and the second component explained 4.1% (Fig 6). In 2018, the variables of first germination count (FC), ability of germination (G), desiccation tolerance (DT) and dry weight (DW) showed a strong correlation with each other, forming a group in the same quadrant as vigor evaluated by t50. The water content (WC) followed the opposite behavior to the aforementioned variables. We emphasize that only in 2017 did seed longevity (p50) correlate with accumulated rainfall (Rainfall) and with degree days (DD) in the late maturation phase at 76 DAF (Fig 6).

In the agricultural year of 2017, there was an additional 12% of thermal sum, about 227 degrees days in absolute values, during the vegetative and reproductive stages of the crop, compared to the agricultural year of 2018. In the year 2017, considering the vegetative and reproductive stages (around 115 days), there was 23% more rain, an increase of 164 mm (Table 1 and S1 Fig). Also, in 2017, the longevity of the peanut seeds reached a maximum value at 76 DAF, the thermal sum of the maternal environment was 26% higher, about 70 degrees days in absolute values (Table 1). In addition, we also observed that the amount of rainfall was lower in the year 2017, in absolute values, in the late maturation phase compared to 2018.

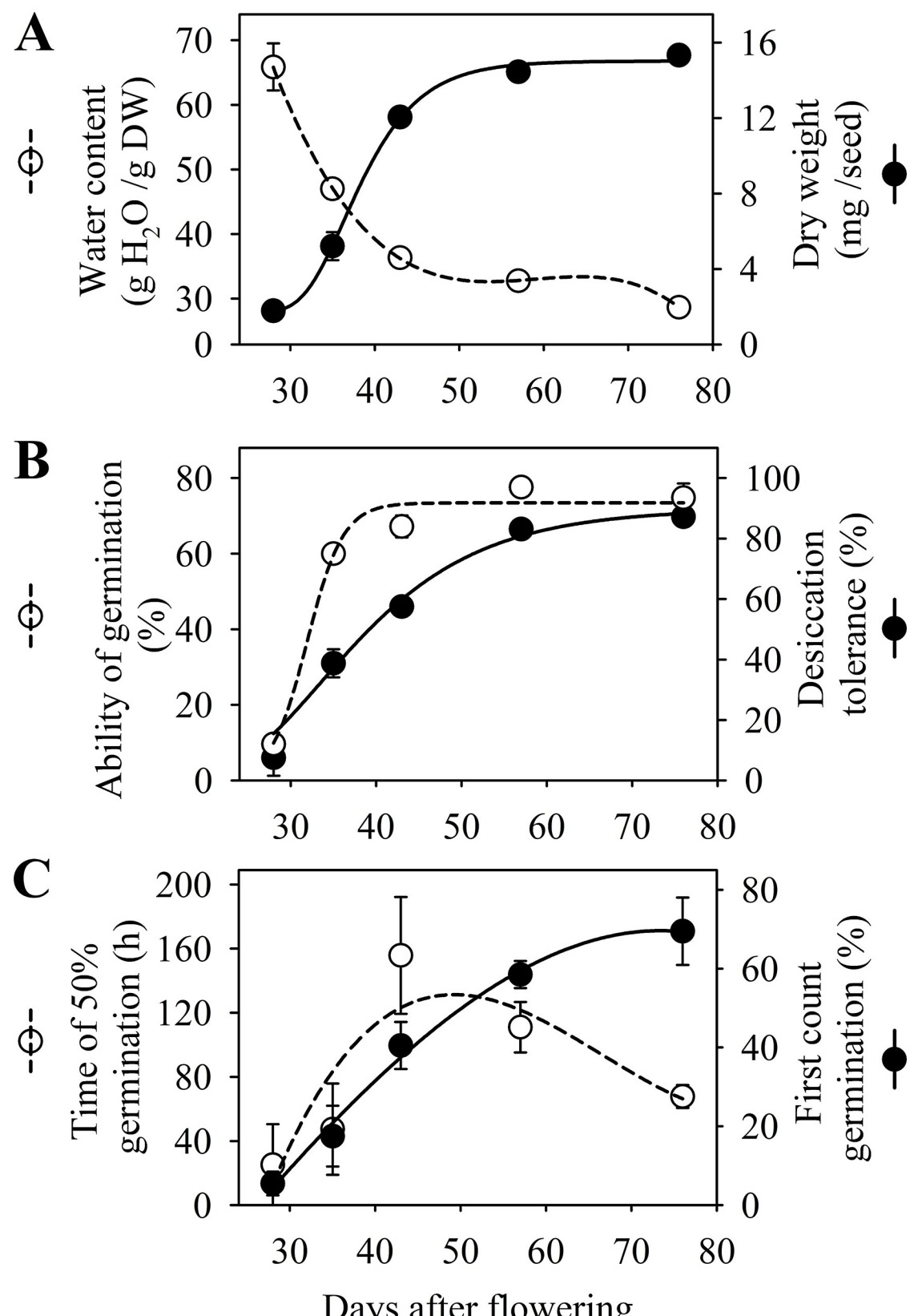

**Fig 2. Physiological characterization of *Arachis hipogaea* L. seed maturation.** (A) Water content and dry weight. (B) Acquisition of germinability and desiccation tolerance. (C) Acquisition of vigor measured at time of 50% ability of germination (t50; hours) and first count of germination. Crop season 2017.

## Discussion

Initially, we present the events associated with the acquisition of physiological quality of peanut seeds (*Arachis hypogaea* L.; Virgínia group) during the maturation and late maturation phases. We verified that even after mass maturity, the seeds continued to acquire quality. This brings new information regarding the period of maturation and late maturation, and the ideal point of harvesting superior quality seeds for this species. Finally, we report the positive influence of the maternal environment, such as rain and thermal sum, in the acquisition of peanut seed longevity.

In the early stages of development, the water content of peanut seeds was high, gradually decreasing until 76 DAF (Figs 2A and 3A). This behavior proved to be similar to what was reported in studies with seeds of other legumes such as *Medicago truncatula* [12] and soybean [9]. The high water content in this initial phase supports the activities of cell division and expansion [18], playing a key role in the synthesis of nucleic acids in parts of the embryo, such as cotyledons and embryonic axis [33]. It also ensures the transfer of dry matter from maternal tissues to the reserve organs of the developing embryo (Figs 2A and 3A), a phenomenon that occurs preferably in an aqueous medium [34]. The gradual dehydration observed (Figs 2A and 3A) results in a decrease in the metabolism of the seed from a limit state of moisture in its tissues, an event reported as part of a natural mechanism in seeds of orthodox species [35].

The maturation (filling) phase of the peanut seeds ended at 57 DAF (Figs 2A and 3A). In Arabidopsis [19] and legumes such as *Medicago truncatula* [12], cowpea [13] and soybean [10], the accumulation of dry weight was reported as an event that increases up to a certain stage of maturation, called mass maturity, remaining stable until the end of seed development. This behavior has also been reported in classical studies with peanut seeds [33, 36], being explained mainly due to the increase in lipid, protein and carbohydrate reserves in the seed organs of this species throughout maturation [37]. It should be noted that lipids and proteins are the major constituents of the chemical composition of peanut seeds, representing around 48% and 31% of reserves, respectively [18]. Therefore, our results showed that until 57 DAF the transfer of dry matter to the embryo increases and then ceases, which characterizes the mass maturity phase (Figs 2A and 3A).

When exposed to desiccation at 28 DAF, the seeds did not express any capacity to germinate. As maturation progressed, the ability of germination was consolidated in parallel by the accumulation of dry weight and the loss of water (Figs 2A, 2B, 3A and 3B). In *Medicago truncatula* [11], soybean [9] and Arabidopsis [19] the desiccation tolerance is fully established before the end of the seed filling. Our results corroborate the behavior reported for these species, considering that the peanut seeds started to tolerate desiccation shortly before mass maturity (Figs 2A, 2B, 3A and 3B). It is believed that during maturation the accumulation of sugars (RFO's) [9] and certain proteins such as LEAs [28] and HSPs [38] play a key role in protecting the seeds from desiccation, ensuring the structural integrity of the macromolecules [35, 39], while the water content decreases (Figs 2B and 3B). Our results support the idea that peanut plants fall into the group of species that produce orthodox seeds, which tolerate desiccation as part of the preparatory mechanisms for the dry state [20, 40].

Following the events described, the ability to germinate and generate seedlings quickly indicated the acquisition of seed vigor up to 76 DAF (Figs 2C and 3C). The progress of this attribute did not necessarily end after the acquisition of mass maturity, similar to what was

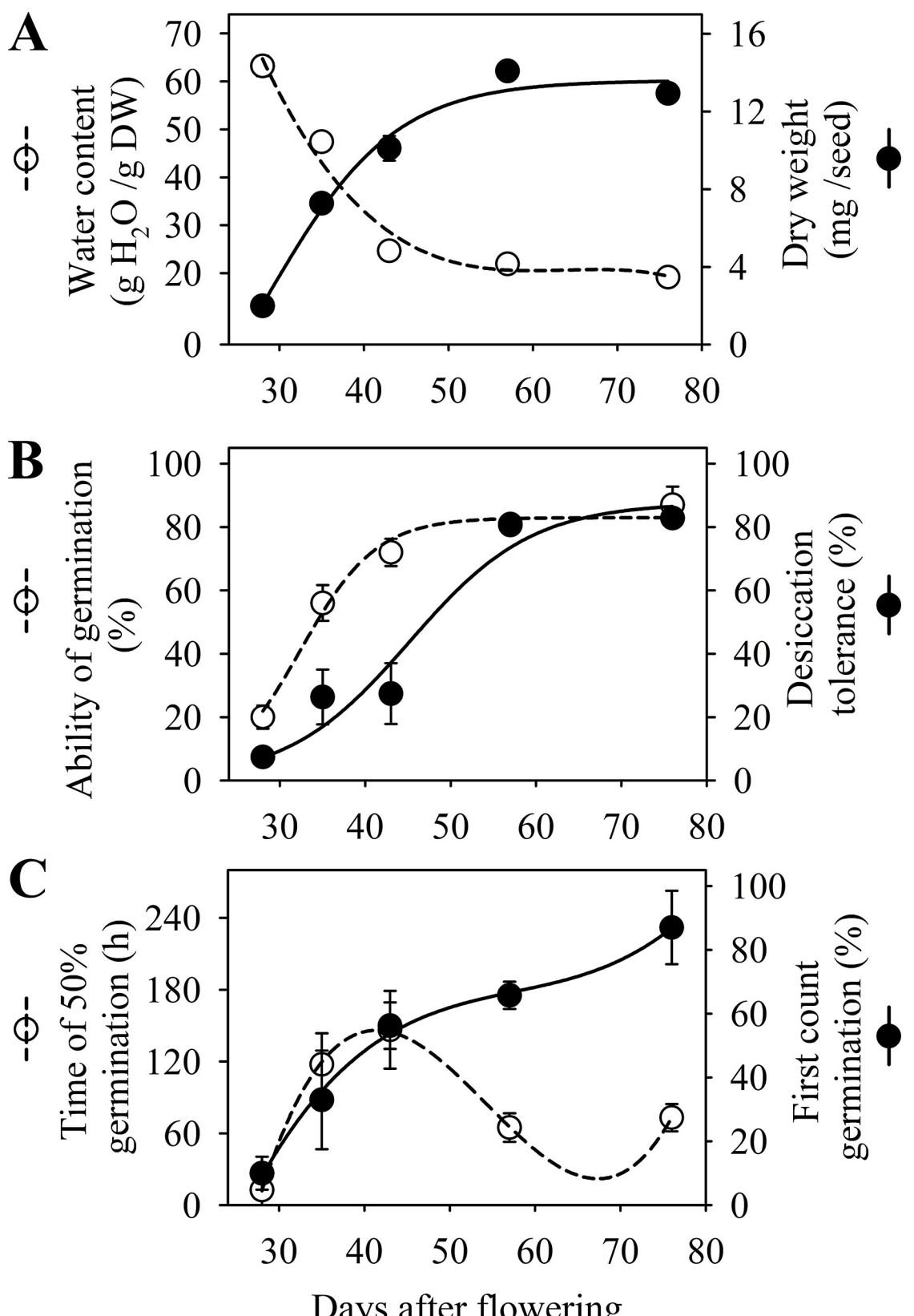

**Fig 3. Physiological characterization of *Arachis hipogaea* L. seed maturation.** (A) Water content and dry weight. (B) Acquisition of germinability and desiccation tolerance. (C) Acquisition of vigor measured at time of 50% ability of germination (t50; hours) and first count of germination. Crop season 2018.

presented in studies performed with pepper seeds [41], okra [22] and soybean [9]. In peanut seeds, the accumulation of phytopherols, a special class of structural lipids, increases during maturation, and they may play a key role in seed performance and seedling vigor [42]. In addition, it is worth noting that the expression of the seed vigor also includes its storage potential [6, 8]. In this regard, peanut seeds from 43 DAF started to show storage potential (35˚C and 75% R.H.) expanding this capacity with the advance of the late maturation phase (Fig 4). This final phase of seed development is decisive for building vigor in legumes such as soybean [10], as we observed for peanut seeds between 43 to 76 DAF (Figs 2C and 3C).

For the acquisition of longevity, the final stage of seed development at 76 DAF also showed higher p50 values compared to the other stages (Fig 5A and 5B). The stabilization of the biological system is described as a result of protection mechanisms, such as the vitreous cytoplasm, formed during the drying of the seeds [35], and the repair mechanism, both of which delay the deterioration reactions favoring the maintenance of viability of the seeds during storage [8]. In this work, we understand that the final stages of maturation, around 25% of the total development period, are crucial for the production of peanut seeds with superior longevity (Fig 5B). In species like *Medicago truncatula*, the seed acquires tolerance to desiccation in the maturation phase, and consequently, longevity in the late maturation phase [38], which we also observed for peanut seeds (Figs 2B, 3B, 5A and 5B). It should be emphasized that longevity

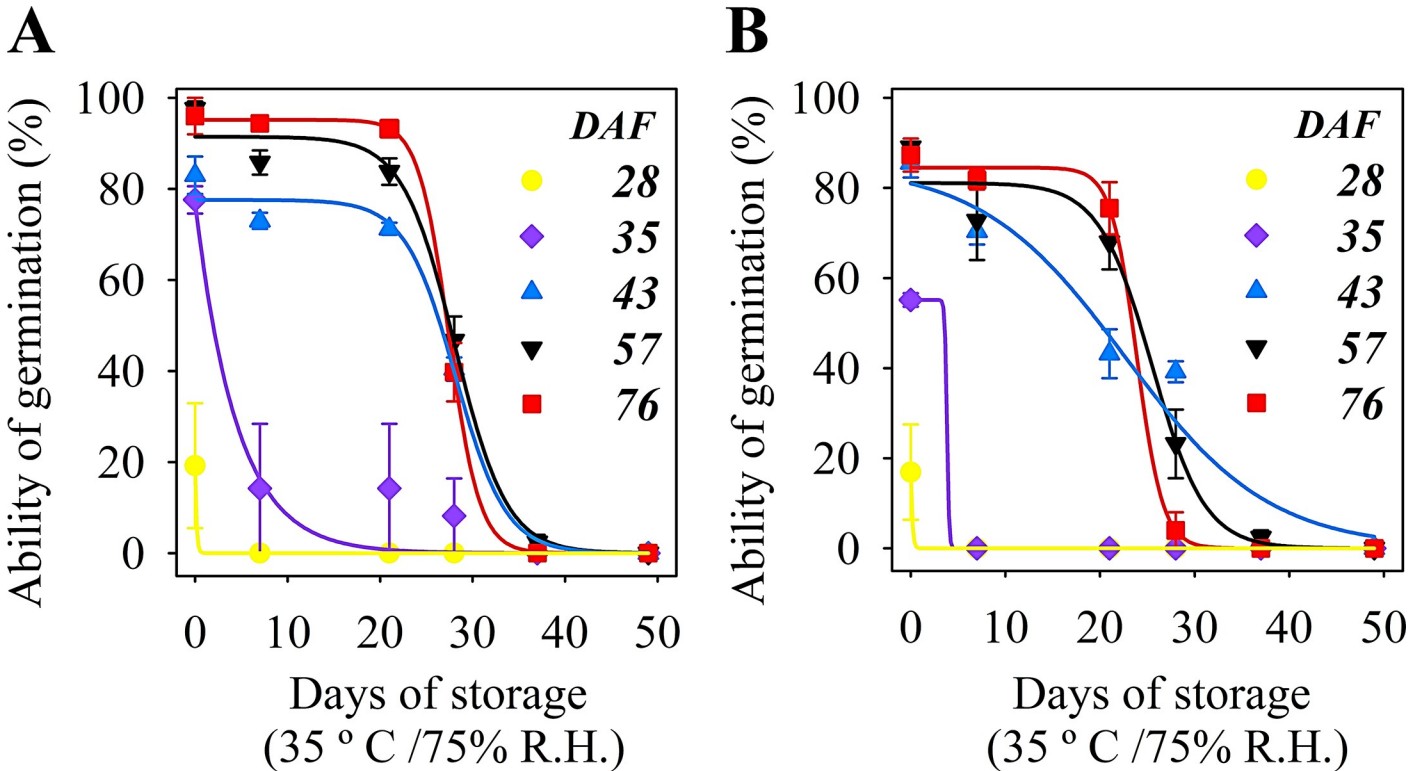

**Fig 4. Loss of the ability of germination of *Arachis hipogaea* L. seeds during storage at 75% RH at 35˚C.** (A) Seeds were harvested at the crop season 2017 during maturation at diferent developing stages. (B) Seeds were harvested at the crop season 2018 during maturation at different developing stages.

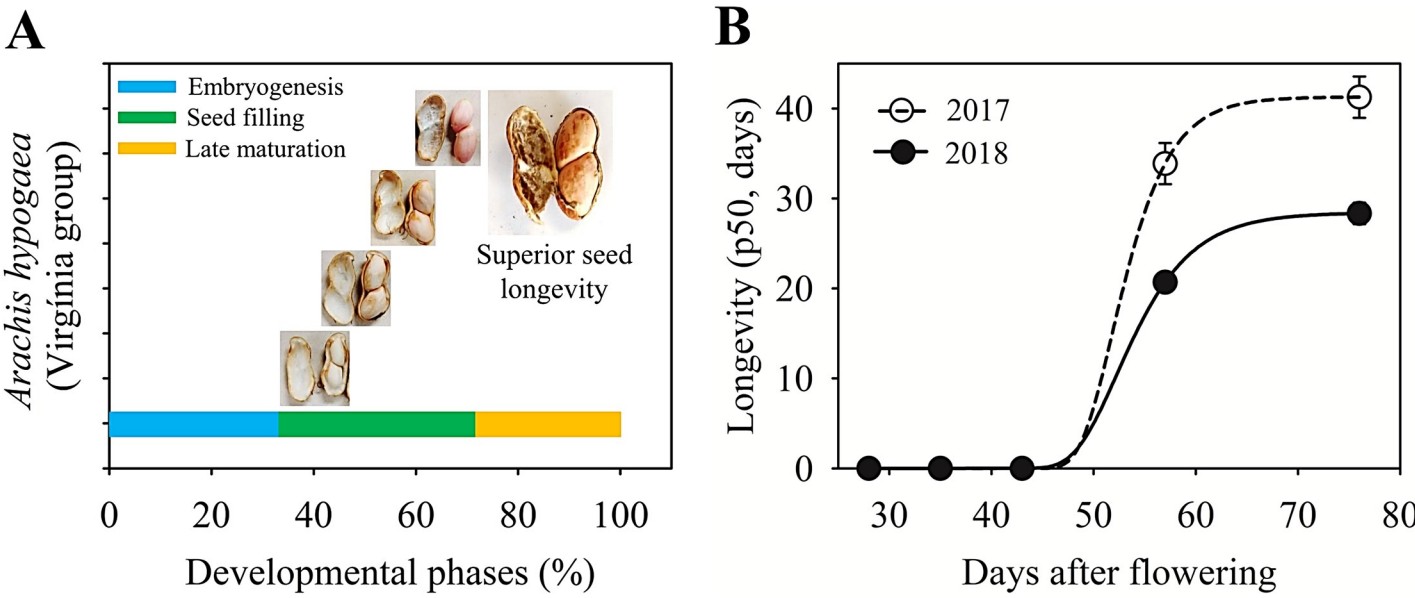

**Fig 5. The relation between the developing stages and the acquisition of longevity in peanut seeds.** (A) Duration of the three developing phases of peanut seed. (B) Acquisition of longevity from the crop season 2017 (white circle) and from the crop season 2018 (black circle), expressed in p50 (period in days that the viability of the seed reduces to 50%).

is the temporal component of seed survival in the dry state [20], and our results, supported by previous studies [9, 12], demonstrated that the maximum physiological quality is acquired in the late maturation phase (Fig 5A and 5B).

We identified that factors in the maternal environment favored the acquisition of seed longevity (Figs 5B and 6). It is believed that in contrast to desiccation tolerance, longevity has high plasticity [20], which means being under strong influence of the environment during seed development [43, 44]. We showed that at 76 DAF in 2017, p50 was correlated with accumulated rain (rainfall) and thermal sum (DD: degrees days) during the peanut cycle (Fig 6). Water availability and temperature during development are crucial for the acquisition of maximum seed longevity [45, 46]. We observed that these factors were different between the crop seasons of the two years studied (Table 1 and S1 Fig). It is known that water deficit conditions accelerate or delay the acquisition of longevity in leguminous seeds [11] interfering in the accumulation of reserves, the ability of germination and vigor of the seedlings [47]. Therefore, we believe that the maternal environment was favorable to the accumulation of reserves and other protection mechanisms that influence the response to longevity acquisition, resulting in the superior expression of this attribute in the seeds produced in the first year of the study (Fig 5B).

This work brought evidence that after mass maturity, the seeds continue to gain physiological quality (Figs 2–5). Regarding this, events that occurred during the post-filling period, including the molecular mechanisms related to the acquisition of longevity, were studied in legume seeds [9, 11, 12]; although for peanuts there is still a knowledge gap. Thus, we can shed light on the idea that the acquisition of quality after mass maturity of peanut seeds (Figs 2–5) comprises an autonomous phenomenon and, therefore, is independent of the connection of the embryo to the maternal tissues (Fig 1). What we bring in this work is the starting point for understanding the maturation physiology of peanut seeds, including the relationship between the acquisition of seed quality and the production environment under tropical climate conditions (Fig 6).

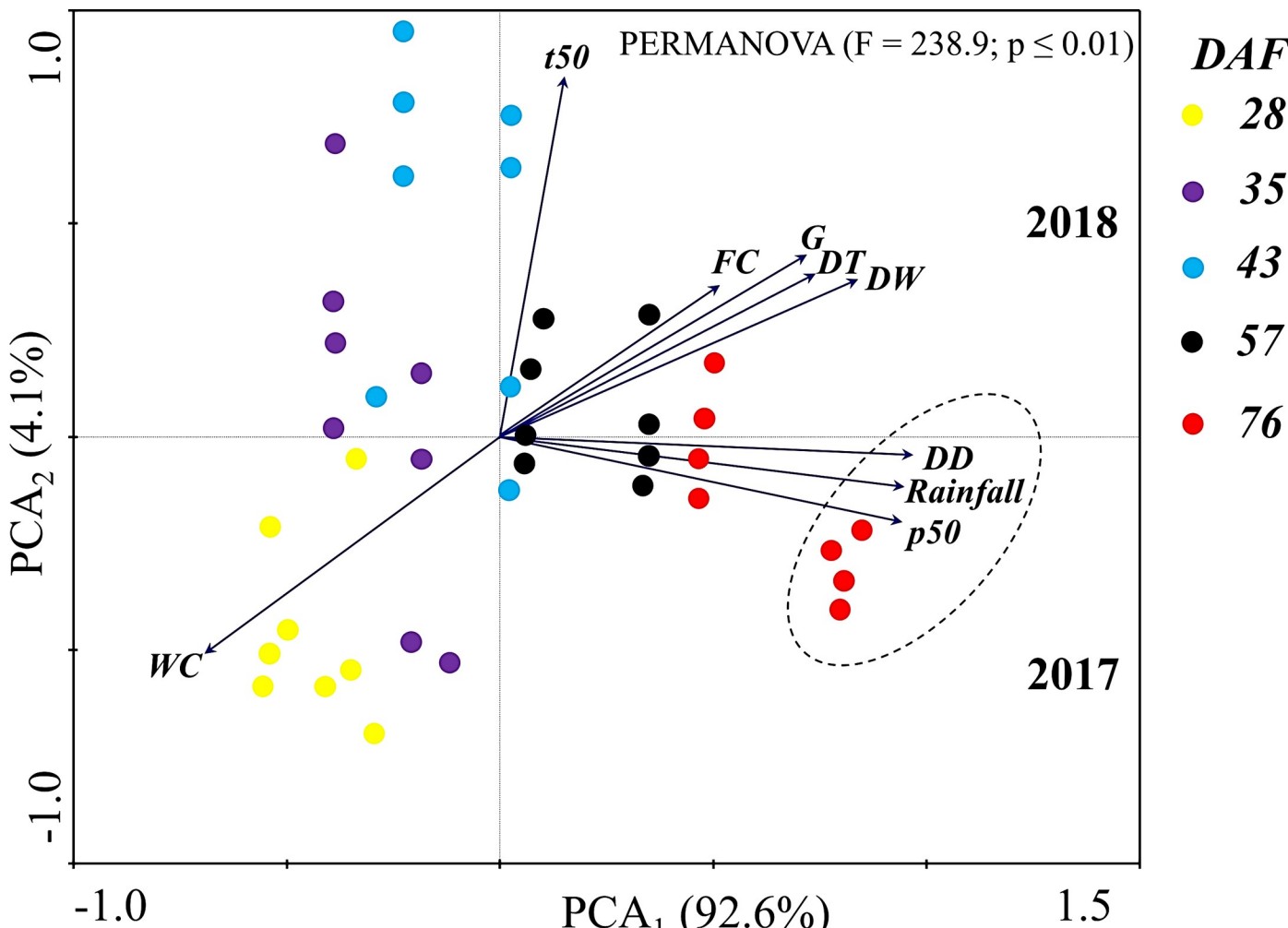

**Fig 6. Analysis of principal components.** The circle indicates the factors that most correlated seed longevity with the maternal environment factors at the 76 DAF (WC: Water content; DW: Dry weight; G: Ability of germination; DT: Desiccation tolerance; t50: Time of 50% germination; FC: First count of germination; p50: Longevity and DD: Degrees days) evaluated in the two crop seasons (2017 and 2018).

**Table 1. Total accumulated values of rainfall and degree days (DD) during the vegetative and reproductive phases of the peanut plants and the absolute differences between the two crop seasons at 28, 35, 43, 57 and 76 days after flowering (DAF).**

| Stages | Rainfall 2017 | Rainfall 2018 | Absolute difference % [*] | DD 2017 | DD 2018 | Absolute difference % [*] |
|---|---|---|---|---|---|---|
| Vegetative | 176 | 93 | 47 | 655 | 578 | 12 |
| Reproductive | 528 | 447 | 15 | 1256 | 1106 | 12 |
| Total | 704 | 540 | 23 | 1911 | 1684 | 12 |
| DAF | | | | | | |
| 28 | 193 | 207 | 7 | 450 | 415 | 3 |
| 35 | 32 | 18 | 44 | 121 | 103 | 15 |
| 43 | 89 | 3 | 97 | 127 | 123 | 3 |
| 57 | 106 | 43 | 59 | 214 | 209 | 2 |
| 76 | 108 | 176 | 39 | 327 | 257 | 26 |

[*] Subtraction between the average values observed in the crop seasons 2017 and 2018.

In general, we showed that the peanut seed acquires physiological quality sequentially with development, and that harvesting at the right time will improve its quality. As reported in previous studies, peanut seeds have less accumulation of reserves in the early stages of development [33, 42], which also reinforces the idea that they do not have the complete constitution of their chemical composition. Therefore, immature seeds are of low physiological quality and have less accumulation of reserves for the production of vigorous seedlings in the field, which harms the formation of crops [17]. Producing highly vigorous seeds is essential for agriculture, as it has an impact on crop productivity through the proper establishment of the plant in optimal and stressful conditions [6]. Thus, the results presented here for seeds of the Virgínia group serve as a basis to achieve this objective. It is worth mentioning that more research is needed for the development of the peanut seed sector. Therefore, studies using other botanical groups (Runner, Valencia and Spanish) may be necessary to improve the understanding of the acquisition of seed quality in *Arachis hypogaea*. This will add fundamental knowledge regarding the production of superior quality seeds. These higher quality seeds undoubtedly will benefit the production and processing industry.

## Conclusions

We demonstrate that the physiological quality of peanut seeds is acquired during the maturation and late maturation phases and that mass maturity does not coincide with maximum seed physiological quality. We emphasize that the late maturation phase is considered the moment for the harvest of peanut seeds aiming at maximum physiological quality. Our findings support the idea that the adequate proportion of rainfall and thermal sum in the maternal environment are determining factors for the production of peanut seeds with superior longevity.

## Supporting information

**S1 Fig. Daily rainfall and maximum and minimum temperatures in the city of Sertãozinho in the State of São Paulo-Brazil during seed production.**
(DOCX)

**S1 Table. Statistical information of the data evaluated in peanut seeds during maturation and late maturation.** Crop season 2017.
(DOCX)

**S2 Table. Statistical information of the data evaluated in peanut seeds during maturation and late maturation.** Crop season 2018.
(DOCX)

**S3 Table. Statistical information on the observed data for peanut seed longevity, as assessed by p50 (sigmoid curves).** Crop season 2017 and 2018.
(DOCX)

## Acknowledgments

We are also grateful to COPERCANA for their support during the experiments, to Thiago Barbosa Batista for his valuable suggestions during the correction of the manuscript and to Mr. Roger Hutchings for the English review of the manuscript.

## Author Contributions

**Conceptualization:** Gustavo Roberto Fonseca de Oliveira, Edvaldo Aparecido Amaral da Silva.

**Data curation:** Maurício Hideki Okada, Gustavo Roberto Fonseca de Oliveira, Maria Márcia Pereira Sartori.

**Formal analysis:** Gustavo Roberto Fonseca de Oliveira.

**Funding acquisition:** Edvaldo Aparecido Amaral da Silva.

**Investigation:** Maurício Hideki Okada.

**Methodology:** Maurício Hideki Okada, Gustavo Roberto Fonseca de Oliveira, Edvaldo Aparecido Amaral da Silva.

**Project administration:** Edvaldo Aparecido Amaral da Silva.

**Resources:** Edvaldo Aparecido Amaral da Silva.

**Supervision:** Edvaldo Aparecido Amaral da Silva.

**Validation:** Gustavo Roberto Fonseca de Oliveira.

**Visualization:** Gustavo Roberto Fonseca de Oliveira, Carlos Alexandre Costa Crusciol.

**Writing – original draft:** Gustavo Roberto Fonseca de Oliveira, Edvaldo Aparecido Amaral da Silva.

**Writing – review & editing:** Gustavo Roberto Fonseca de Oliveira, João Nakagawa, Edvaldo Aparecido Amaral da Silva.

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
