## [Decision Letter · Decision Letter 0]

5 Mar 2021

PONE-D-21-02258

Acquisition of the physiological quality of peanut (Arachis hypogaea L.) seeds during maturation under the influence of the maternal environment"

PLOS ONE

Dear Dr. Amaral da Silva,

Thank you for submitting your manuscript to PLOS ONE. After careful consideration, we feel that it has merit but does not fully meet PLOS ONE’s publication criteria as it currently stands. Therefore, we invite you to submit a revised version of the manuscript that addresses the points raised during the review process.

We look forward to receiving your revised manuscript.

Kind regards,

Craig Eliot Coleman, PhD

Academic Editor

PLOS ONE

Journal Requirements:

2. Please include your tables as part of your main manuscript and remove the individual files. Please note that supplementary tables (should remain/ be uploaded) as separate "supporting information" files

Reviewers' comments:

Reviewer's Responses to Questions

**Comments to the Author**

1. Is the manuscript technically sound, and do the data support the conclusions?

Reviewer #1: Yes

2. Has the statistical analysis been performed appropriately and rigorously? 

Reviewer #1: Yes

3. Have the authors made all data underlying the findings in their manuscript fully available?

Reviewer #1: Yes

4. Is the manuscript presented in an intelligible fashion and written in standard English?

Reviewer #1: Yes

5. Review Comments to the Author

Reviewer #1: This manuscript was well written, reporting the physiological quality during seed maturation. The experiment was well designed. The authors traced the seed development from flowering to late-stage maturation by labelling individual flowers and pods under different environment conditions. After harvesting single seeds, the physiological quality was evaluated by six traits (water content, dry weight, ability to germination, desiccation tolerance, and vigor and longevity). Based on the evaluation results, the authors recommended that the best time to harvest for physiology quality was between 57-76 days after flowering in the late stage of maturation. The information presented here is useful for peanut farmers and researchers, but I have the following concerns. To enhance the quality of this manuscript, I have the following comments, and suggestions.

Major revisions:

1. The term of germination used here is not accurate and confusing. It should be the ability of germination. It should be changed throughout the text and figures.

2. In the discussion, the authors should elaborate how the physiological quality can affect peanut seed yield, seed nutritional quality, and seed processing quality except seed harvesting time and seed ability of germination.

3. There are several botanical types. Here the authors only evaluated Virginia type. They should discuss the limitation their results for other botanical types.

4. In addition, the authors should also give the information of seed dormancy for Virginia type or this cultivar if possible, because seed dormancy can significantly affect these traits investigated.

Minor revisions:

Page 11 line 123, misspelling “Thay”

Page 12 line 133, misspelling “parcentage”

List reference for calculation of longevity on page 12.

6. PLOS authors have the option to publish the peer review history of their article (what does this mean?). If published, this will include your full peer review and any attached files.

Reviewer #1: No

---

## [Author Response · Author response to Decision Letter 0]

31 Mar 2021

Manuscript Number: PONE-D-21-02258

Acquisition of the physiological quality of peanut (Arachis hypogaea L.) seeds during maturation under the influence of the maternal environment

Craig Eliot Coleman PhD,

Academic Editor of PLOS ONE

 Dear Dr. Coleman,

We thank you and the reviewer for the corrections and suggestions made in the manuscript. Below, we provide our answers point-by-point to the comments and suggestions made.

With warm regards,

Amaral da Silva (corresponding author on behalf of co-authors)

Reviewer #1: 

This manuscript was well written, reporting the physiological quality during seed maturation. The experiment was well designed. The authors traced the seed development from flowering to late-stage maturation by labelling individual flowers and pods under different environment conditions. After harvesting single seeds, the physiological quality was evaluated by six traits (water content, dry weight, ability to germination, desiccation tolerance, and vigor and longevity). Based on the evaluation results, the authors recommended that the best time to harvest for physiology quality was between 57-76 days after flowering in the late stage of maturation. The information presented here is useful for peanut farmers and researchers, but I have the following concerns. To enhance the quality of this manuscript, I have the following comments, and suggestions.

Answer: We improved all the points that the reviewer deemed necessary. Below we answered all the proposed questions point by point. All changes are marked in the main text. We hope to have met the expectations. We agree that the manuscript has improved after the corrections and suggestions made by the reviewer.

Detailed comments

Major revisions

1. The term of germination used here is not accurate and confusing. It should be the ability of germination. It should be changed throughout the text and figures.

Answer: We added the term "ability of germination" throughout the text (Lines 22, 56, 110, 141, 176, 192, 234 and 278) and also in the tables and figures (Lines 458, 463, 466 and 478) in the manuscript and in the supporting information (Lines 17 and 24).

2. In the discussion, the authors should elaborate how the physiological quality can affect peanut seed yield, seed nutritional quality, and seed processing quality except seed harvesting time and seed ability of germination. 

Answer: We have prepared a final paragraph (Lines 292 to 305) highlighting the importance of physiological seed quality for peanut yield. We also reinforced the need to harvest seeds at the moment of maximum seed quality, shedding light on the idea that the constitution of reserves has an influence on the vigor of seedlings and the formation of the crop. Finally, we commented on the importance for growers and to the seed to industry to harvest seeds with high quality. 

3. There are several botanical types. Here the authors only evaluated Virginia type. They should discuss the limitation their results for other botanical types.

Answer: In the last paragraph of the manuscript, we also highlight that our work with the Virginia group gathers information that provides the basis for further research. We emphasize that it is necessary to study other botanical groups, in order to consistently elucidate the events occurring during the maturation and late maturation for the species (Arachis hypogaea).

4. In addition, the authors should also give the information of seed dormancy for Virginia type or this cultivar if possible, because seed dormancy can significantly affect these traits investigated.

Answer: Thanks for the great suggestion. We added in the "Material and Methods" section (Lines 119 and 120) the information of how we broke any residual dormancy in peanut seeds. For this, we used a sodium hypochlorite solution recommended for this purpose. We adapted this information for peanut seeds and we had no dormancy problems during the experiments. The reference from which we extracted this information was added to the list of references in the manuscript (number 27) as shown below.

27. Mahajan G, Mutti NK, Jha P, Walsh M, Chauhan BS. Evaluation of dormancy breaking methods for enhanced germination in four biotypes of Brassica tournefortii. Scientific Reports. 2018; 8: 1-8. doi: 10.1038/s41598-018-35574-2. 

Minor revisions

1. Page 11 line 123, misspelling “Thay”

Answer: Thanks. We performed the correction as suggested (Line 125)

2. Page 12 line 133, misspelling “parcentage”

Answer: Thanks. We performed the correction as suggested (Line 135).

3. List reference for calculation of longevity on page 12.

Answer: Thanks. We presented more details regarding how the calculation of p50 (longevity) was performed from sigmoid survival curves using germination data during the storage of peanut seeds (Lines 142 to 144). The article containing the information used (number 30) was listed in the item "References" and described below. 

30. Ellis RH, Roberts EH. Improved equations for the prediction of seed longevity. Annals of Botany. 1980; 45: 13-30. doi: 10.1093/oxfordjournals.aob.a085797.

Other modifications

1. In the item "Acknowledgments" we acknowledge the researcher who contributed to the final version of this manuscript for his innovative ideas and suggestions (Lines 319 and 320)

2. We added two more references in the manuscript (number 27 and 30). Therefore, we changed the numerical order of the references presented in the manuscript; which was also revised according to the standards of PLOS ONE.

---

## [Editor Report · Decision Letter 1]

5 Apr 2021

Acquisition of the physiological quality of peanut (Arachis hypogaea L.) seeds during maturation under the influence of the maternal environment

PONE-D-21-02258R1

Dear Dr. Amaral da Silva,

We’re pleased to inform you that your manuscript has been judged scientifically suitable for publication and will be formally accepted for publication once it meets all outstanding technical requirements.

Kind regards,

Craig Eliot Coleman, PhD

Academic Editor

PLOS ONE
---

## [Editor Report · Acceptance letter]

23 Apr 2021

PONE-D-21-02258R1 

Acquisition of the physiological quality of peanut (*Arachis hypogaea* L.) seeds during maturation under the influence of the maternal environment 

Dear Dr. Amaral da Silva:

I'm pleased to inform you that your manuscript has been deemed suitable for publication in PLOS ONE. Congratulations! Your manuscript is now with our production department. 

Kind regards, 

on behalf of

Dr. Craig Eliot Coleman 

Academic Editor

PLOS ONE